# Enhanced Thermal Transport Properties of Graphene/SiC Heterostructures on Nuclear Reactor Cladding Material: A Molecular Dynamics Insight

**DOI:** 10.3390/nano12060894

**Published:** 2022-03-08

**Authors:** Lei Wu, Xiangyang Sun, Feng Gong, Junyi Luo, Chunyu Yin, Zhipeng Sun, Rui Xiao

**Affiliations:** 1Key Laboratory of Energy Thermal Conversion and Control of Ministry of Education, School of Energy and Environment, Southeast University, Nanjing 210096, China; npicwulei@163.com (L.W.); 18365096312@163.com (X.S.); luojunyi1010@163.com (J.L.); 2Science and Technology on Reactor System Design Technology Laboratory, Nuclear Power Institute of China (NPIC), Chengdu 610041, China; yincy909@163.com (C.Y.); superszp@163.com (Z.S.)

**Keywords:** thermal conductivity, graphene/SiC heterostructure, molecular dynamics simulation, cladding material, nuclear reactor

## Abstract

Owing to the excellent thermal properties of graphene, silicon carbide (SiC) combined with graphene is expected to obtain more outstanding thermal performance and structural stability at high temperatures. Herein, the thermal conductivity of graphene/SiC heterostructures (GS-Hs) with different structures and atomic orientations was calculated through non-equilibrium molecular dynamics (NEMD) simulations. The temperature dependence and size effect on the thermal transport properties of GS-Hs were systematically investigated and discussed. The continuous addition of graphene layers did not always have a positive effect. The thermal transport performance of GS-Hs approached the intrinsic thermal conductivity of SiC when the interaction gradually decreased with the distance between SiC and graphene. Studies on temperature and size dependence show opposite trends. The enhancement effect of graphene was limited at small distances. The thermal conductivity of GS-Hs had a negative correlation with temperature and increased with the system size. Meanwhile, the thermal conductivity of GS-Hs was predicted to be 156.25 (W·m^−1^·K^−1^) at the macroscopic scale via extrapolation. The model established in this paper is also applicable to other material simulation processes, as long as the corresponding parameters and potential functions are available. This study will provide inspiration for the optimized design and preparation of highly efficient cladding materials in nuclear reactors.

## 1. Introduction

Similar to solar, hydro, wind and other renewable energy sources, nuclear energy is a low-carbon and environmentally friendly energy source [1,2]. Compared with conventional fossil fuels, nuclear power generation does not encounter the risk of resource depletion and environmental pollution [3]. At present, pressurized water reactors (PWRs) are the main type of nuclear power plants in commercial operation worldwide [4]. As the most crucial part of the reactor for power generation, the reactor core needs high-performance cladding material for protection [5,6]. Silicon carbide (SiC) is expected to replace traditional zirconium-based alloys and becomes a new type of high-efficiency nuclear fuel cladding material because of its high strength, high thermal conductivity and excellent thermal and irradiation stability [7]. Despite the outstanding thermal conductivity of commercial SiC materials, the industry wishes to further improve the thermal performance of SiC-based cladding materials in view of the extension of the refueling cycle, the enhancement of safety margins and further applications in nuclear power plants [8].

In recent years, graphene has attracted extensive attention both from the scientific community and the industry, owing to its unique thermal properties, showing huge potential to address the thermal adjustment challenge in material design [9]. Balandin et al. [10] firstly demonstrated that the thermal conductivity of graphene is up to 2000 W·m^−1^·K^−1^ by the measurement of single-layer graphene using the photo Raman technique. In the application field of graphene, it is often used as an additive for composite materials to improve thermal transport properties [11]. For example, Gong et al. [12] built graphene/MoS_2_ heterostructures to enhance the electrochemical and thermal transport properties of MoS_2_. Chu et al. [13] reported that the combination of Cu and graphene nanosheets by using the vacuum filtration method would contribute to the efficient performance of in-plane heat dissipation. Shen et al. [14] studied graphene/epoxy composites by adding multilayer graphene sheets to improve thermal conductivity. Although there are numerous studies on the thermal transport properties of graphene-based composite materials in other fields, few studies and theoretical analyses have addressed the thermal conductivity of graphene as a nuclear reactor cladding material. 

In addition, the study of the composite of SiC and graphene has also received some attention in recent years. Cheng et al. [15] synthesized SiC nanowires with graphene aerogel via chemical vapor osmosis to obtain an extremely high performance, electromagnetic-absorbing composite material. Vajdi et al. [16] investigated the effect of adding graphene nanosheets to TiB_2_-SiC via discharge plasma sintering on the microstructure and thermal conductivity of the composite material. In order to improve the fracture toughness of Si_3_N_4_/SiC ceramics, Yang et al. [17] added 0.3% graphene to the material and studied the microstructure, mechanical properties and toughening mechanism.

Herein, we propose to construct a graphene/SiC heterojunction structure (GS-Hs) to obtain an effective thermal conductive composite. The great thermal conductivity and thermal stability of graphene are expected to further improve the thermal properties of SiC and thus to achieve excellent structural stability and safety reliability under high-temperature conditions. In this work, we quantitatively and systematically investigated the thermal transport properties of GS-Hs with different structures and atomic orientations via the non-equilibrium molecular dynamics (NEMD) method. The effects of various numbers of graphene and SiC layers on the thermal properties of heterojunction structures were investigated using the pair potential (Lennard-Jones and Airebo) and many-body potential (Tersoff) models. In addition, the influences of temperature and simulation system size on the heat transport performance of GS-Hs were studied. Providing valuable suggestions toward optimizing the effective thermal transport properties in the novel SiC composite material, the modeling of the thermal performance of GS-Hs accomplished in this study will inspire the better design of cladding materials in nuclear reactors.

## 2. Computational Methods

### 2.1. LAMMPS Calculation

All the simulations were conducted using Large-scale Atomic/Molecular Massively Parallel Simulator (LAMMPS), which is an open-source code developed by Sandia National Laboratory in America [18]. LAMMPS can study millions of atomic and molecular systems in a variety of ensembles, including gaseous, liquid and solid phases, and supports multiple potential functions with excellent parallel scalability [19]. 

In order to investigate the effects of different structure of GS-Hs, we performed NEMD simulations using LAMMPS. Kawamura et al. [20] studied the thermal conductivities of different phases of SiC and demonstrated that 3C-SiC has the largest thermal conductivity owing to a unique isotropic structure compared with other polytypes (2H-, 4H- and 6H-). Therefore, 3C-SiC was selected as the substrate in GS-Hs. The lattice constants of 3C-SiC and the graphene sheet were 4.348 Å and 2.46 Å, respectively. Prior to calculation, GS-H heterojunctions were built. As shown in Figure 1a–c, the GS-Hs was formed by positioning the graphene layer (1 × 7) together with the SiC layer (1 × 4), and the heterostructure had a lattice mismatch of less than 1%. Due to the fact that adjacent graphene planes within a graphite crystal are connected by weak van der Waals interactions (about 50 meV) with a spacing of 3.4 Å, the distance between graphene and SiC monolayers was set as 3.4 Å in the GS-Hs, as illustrated in Figure 1d [21,22]. In all simulations, periodic boundary conditions were employed in x, y and z directions [23,24].

### 2.2. Potential Function

In molecular dynamics (MD) simulation, the potential function is primally used to describe the interaction between atoms (molecules) [25]. In this study, Tersoff and Airebo were selected to obtain the interaction between the atoms in the respective layers of SiC and graphene, while the interlayer interaction between SiC layer and graphene was calculated using Lennard-Jones (LJ) potential. Tersoff potential is the most frequently used for SiC among the many-body empirical potentials and the parameters have been modified or reparametrized to improve the accuracy by many researchers [26]. The function potentials most widely used for graphene include the Tersoff, Rebo and Airebo potentials. Airebo can be visualized as the sum of Rebo potential, LJ potential and torsional term [27]. Due to the fact that there is some uncertainty in the intensity and form of the interactions between the graphene atoms and the SiC atoms, the LJ potential function is employed to express the interaction energy between Si-C, C-C and C-Si atoms in different layers [28]. The L-J parameters used in the simulations are summarized in Table 1, where ε is the energy parameter and σ is the distance parameter.

### 2.3. NEMD Method

In NEMD simulation, the extensive linear response theory is used to calculate the thermal conductivity by using the system nonequilibrium response caused by external disturbance [29]. The individual atoms are treated as ideal mass points without volume [25]. Classical mechanics indicates that macroscopic variables can be derived by the numerical integration of coordination and velocities between atoms [30]. As shown in Figure 2, the thermal conductivity of GS-Hs was determined by introducing constant heat flux within the heterostructures. Constant heat flux was added into the heat source region in each time step, and the same amount of energy was simultaneously removed from the cool source region. The system was first stabilized at a constant temperature. A time step of 0.5 fs was applied in the calculation. Prior to introducing the constant heat flux, the conjugate gradient minimization scheme was employed to optimize the initial system to a thermal stable state. Subsequently, the system was equilibrated at 300 K for 100,000 time steps under NPT (constant mass, pressure and temperature), and repetitive operation occurred under NVT (constant mass, volume and temperature) successively. Then, the system was switched to the NVE (constant mass, volume and energy) ensemble to eliminate the influence of removing the thermostat in the end [31]. Owing to the applied temperature gradient between the heat source and cool source, a nonlinear effect was observed near the hot and cold regions, producing a strong scattering phenomenon [32].

When the system reached the steady state, the thermal conductivity of finite length L was calculated according to Fourier’s law by measuring the heat flux (flux) and temperature gradient. Fourier’s law of thermal conductivity states that in a given material, there is a positive relationship between the heat flux density and the corresponding temperature gradient, and the proportional coefficient is the thermal conductivity [20]. Fourier’s law can be expressed as:(1)$$J_{y}=-\kappa \frac{dT}{dy}$$
where *J_y_* is one component of thermal current, *dT/dy* is the temperature gradient along the y direction and *κ* is the thermal conductivity. Because of the specific composition surface, the thermal conductivity in the y direction was selected as the calculation of thermal conductivity. The temperature distribution can be simulated by the traditional NEMD at a certain temperature. However, due to the large difference between the structure scale of the computable system and the actual material, the calculated results of thermal conductivity will change with the increase in the simulation scale when the scale is close to or less than the mean free path of crystal phonons [33]. The relationship between the simulated value of thermal conductivity and the length of heat flux transfer direction is as follows:(2)$$\frac{1}{\kappa }=\frac{a^{3}}{4k_{B}\nu }\left(\frac{1}{l_{\infty }}+\frac{4}{L_{b}}\right)$$
where *α* is the lattice constant; *ν* is the group velocity of phonons; *l*_∞_ is the mean free path of phonons; and *L_b_* is the length of heat flow transfer direction. By changing the scale size of the simulated structure, the influence of size effect on the thermal conductivity was explored. By analyzing the above equation, it can be concluded that: 1/*κ* and 1/*L_b_* present an approximate linear relationship. Furthermore, the thermal conductivity at the microscopic size was processed by least squares to draw a linear image. When *L_b_* extended to the macroscopic size, 1/*L_b_* approached 0. The value of 1/*κ* was equal to the intercept of the vertical axis, which was the reciprocal of the actual thermal conductivity.

## 3. Results and Discussion

### 3.1. Composite Structure

The thermal conductivity of GS-Hs with different composite structures was calculated at 300 K, as shown in Figure 3. Different fabrication methods and atomic orientations of GS-Hs determined the atomic interaction and then led to the change in thermal conductivity. Figure 3 demonstrates that the graphene layer had a significant effect on the thermal conductivity of GS-HS. When the thickness of SiC increased gradually, the enhancement of graphene was trapped in simulation conditions. Parameters were calculated using the Lorentz–Berthelot mixed rule [34]. The cut-off distance of LJ potential function for Si-C interaction was 2.5 σ or 8.315 Å [28]. The interaction was removed when the distance between the atoms exceeded 8.315 Å, and the composite material performance approached the intrinsic thermal conductivity value of SiC. When one layer of graphene was introduced into one layer of SiC, the structure was named Gr-1-SiC-1, and it was also applicable to other number of layers [35,36].

As shown in Figure 3a, the distance between graphene carbon atoms and the top atoms of SiC was 7.111 Å in Gr-1-SiC-1, which was less than the cut-off distance. However, the distance between graphene carbon atoms and the bottom atom of SiC was 11.526 Å in the Gr-1-SiC-2, which was larger than the interaction distance of carbon atoms, and the atoms beyond the truncation distance retained their eigenvalues. Therefore, when the thickness of SiC layer increased, the effect of graphene was not obvious, and the value of thermal conductivity approached the inherent attribute of the SiC material. As the number of graphene layers gradually increased, the thermal conductivity of the heterojunction structure was gradually enhanced. When the distance between the graphene and the top SiC atom was 10.606 Å in Gr-3-SiC-1, it exceeded the cut-off distance to reduce the internal interaction.

In addition, the effect of interatomic interactions between graphene and SiC was studied on Si-terminated surfaces. Figure 3b demonstrates that the thermal conductivity followed a similar trend when the atom orientation changed, but the corresponding value was slightly smaller.

### 3.2. Temperature Dependance of Thermal Conductivity

Different substrate temperatures (T = 300, 400…1500, and 1800, 2100…3000 K) were analyzed in this study. Temperature in NEMD simulations is typically calculated based on the mean kinetic energy of the system [37]. The temperature distribution and temperature gradient at different substrate temperatures are shown in Figure 4. It can be concluded that the temperature distribution in GS-Hs had an obvious linear trend along the heat flow at low temperature. As the temperature gradually rose, this trend became unstable with a few fluctuations. This phenomenon may be caused by the fact that the structure of a heterojunction becomes fragile at high temperatures, and the molecular movement becomes more violent and disorderly [38]. Therefore, the adjacent blocks in the calculation model of high temperature display violent differences, presenting such a disordered state.

The thermal conductivity of GS-Hs monotonically decreased with increasing temperature as shown in Figure 5. The temperature dependence of the GS-Hs was consistent with other previous simulations, in which a negative tendency was also shown [37,39]. The relationship with temperature demonstrates that the inelastic scattering of phonons plays a significant role in thermal diffusion. The violent lattice vibration resulted in a further reduction in the phonon thermal conductivity as the temperature increased. Meanwhile, the phonon scattering at grain boundaries had a negative impact on the heat transfer process [40]. 

### 3.3. Finite-Size Effect of Thermal Conductivity

In order to investigate the finite-size effect of thermal conductivity, systems with different simulation lengths (20, 40, 60…200 Å with increment of 20 Å) were studied, and the results are presented in Figure 6. The obvious temperature distribution fluctuation can be observed with the small size of simulated system, especially in the structure of 20 Å. The degree of linear fitting became consistent as the size increased, indicating that the temperature fluctuations along the heat flux became smooth. The phonon mean free path reduction was caused by the finite size. The biggest size of the simulation system in this study that could be achieved was 20 nm, which was much smaller than the mean free path of graphene, leading to the size effect on the calculation results of thermal conductivity [10]. Meanwhile, due to the periodic boundary conditions, the phonons that reached the boundary surface passed through another identical system [33]. At the microscopic scale, the increase in phonon scattering and grain boundary density will greatly reduce the overall thermal conductivity of composite materials. Moreover, the linear temperature gradient between heat source and cool source presents the thermal equilibrium in a simulation system. The thermal conductivity can be calculated by Equation (1) with the temperature gradient.

However, the values of thermal conductivity for simulated systems containing a finite number of atoms were significantly different from the actual values, indicating that the thermal conductivity of these systems are affected by the finite size. For the biggest simulation system, the finite-size effect had crucial influence on the thermal conduction, even with the periodic boundary conditions [41]. Thus, in order to meet the calculation requirements, some systems with as many atoms as possible were constructed. Meanwhile, the approximate actual thermal conductivity value can be inferred by Equation (2). Figure 7a displays that the thermal conductivities of GS-Hs as a function of the system size with a positive correlation. In fact, the relationship should be positive and gradually smooth towards a customization [42]. According to the corollary of Section 2.3, Figure 7b could be produced with simple operations. The thermal conductivity of GS-Hs with the macro dimension was obtained by extrapolating the regression line to the infinite system size. The intercept of the regression line from the coordinate axis was 0.0064, which meant that 156.25 (W·m^−1^·K^−1^) was the actual thermal conductivity of the composite. Some predictable difficulties can be encountered in the extrapolated process at a finite size system. Even though NEMD is widely used to calculate microscale thermal conductivity, its results sometimes serve as references [33].

## 4. Conclusions

In this work, the thermal transport properties of different composite structure GS-Hs were systematically investigated using NEMD simulations and potential functions (Lennard-Jones, Airebo and Tersoff). The influences of the interatomic interaction, temperature and simulation size were investigated systematically. It was found that interatomic interaction has a significant effect on the thermal conductivity of heterojunctions. The interaction was removed when the inter-layer distance exceeded the cut-off distance and the thermal transport performance of the heterojunction approached the intrinsic value of SiC. However, since the application of SiC in nuclear reactors is still immature, we mainly focused on the thermal properties of materials. The evaluation standard is simplified to be the enhancement of thermal conductivity. The temperature dependence of the GS-Hs shows a negative tendency, and the inelastic scattering of phonons plays a significant role in thermal diffusion. The study of size effects demonstrated that the thermal conductivities of GS-Hs are a function of the system size with a positive correlation. The thermal conductivity of 156.25 (W·m^−1^·K^−1^) was predicted for the heterojunctions at the macro-scale. It is valuable to introduce carbon materials such as graphene into the design of cladding tubes, which can be used as references to improve performance. It was determined that the thermal properties of GS-Hs depend on the interatomic interaction, temperature and simulation size based on the simulation results in this research. The computational model can also be applied to the design and conception of other materials. This quantitative study reveals the thermal transport phenomenon and limitations in graphene/SiC heterojunctions, which may guide the design of highly efficient cladding materials for different nuclear energy plants. 

## Figures and Tables

**Figure 1 nanomaterials-12-00894-f001:**
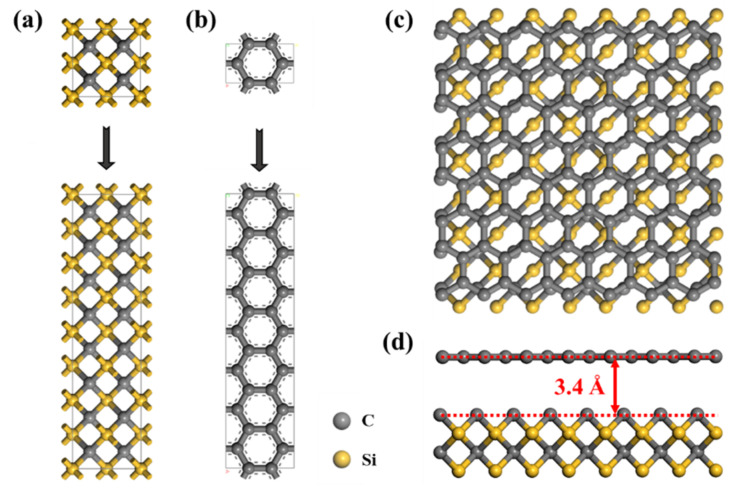
Top views of SiC (**a**), graphene (**b**) and GS-Hs (**c**). The interlayer distance between graphene and SiC is 3.4 Å (**d**). The yellow and gray balls represent Si and C atoms, respectively.

**Figure 2 nanomaterials-12-00894-f002:**
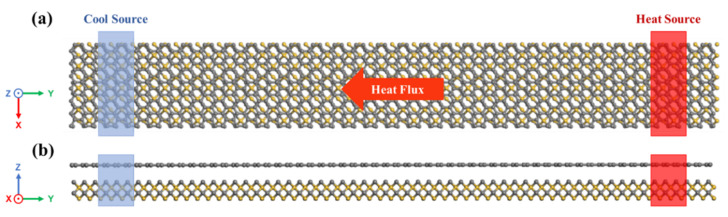
A typical schematic illustration of the NEMD simulation setup. Top (**a**) and side (**b**) views of GS-Hs. Heat flows from the heat source to the cool source, as indicated by the arrow pointing towards the cool source.

**Figure 3 nanomaterials-12-00894-f003:**
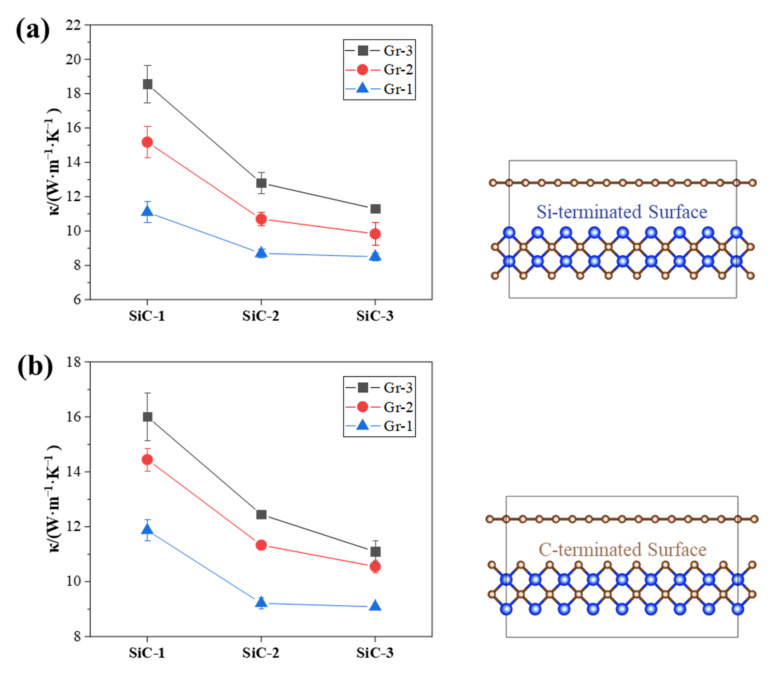
Thermal conductivities and schematic structure of GS-Hs on (**a**) C-terminated surface and (**b**) Si-terminated surface. The error bars are the standard deviation obtained from 3 separate calculations.

**Figure 4 nanomaterials-12-00894-f004:**
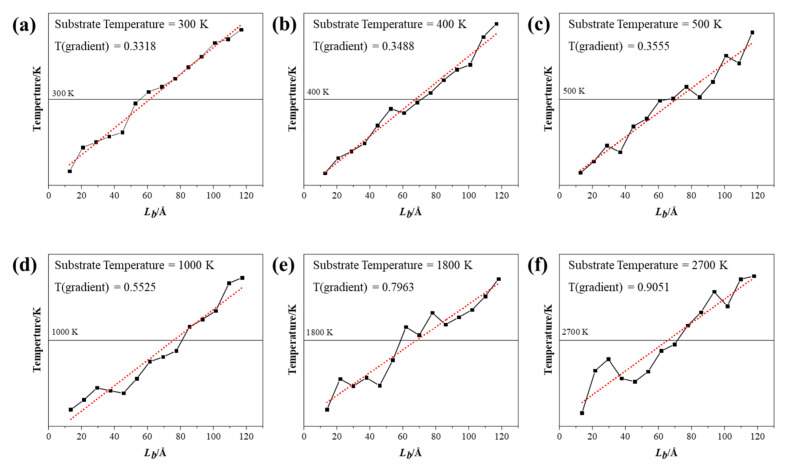
The temperature distribution and temperature gradient at (**a**) 300 K, (**b**) 400 K, (**c**) 500 K, (**d**) 1000 K, (**e**) 1800 K and (**f**) 2700 K.

**Figure 5 nanomaterials-12-00894-f005:**
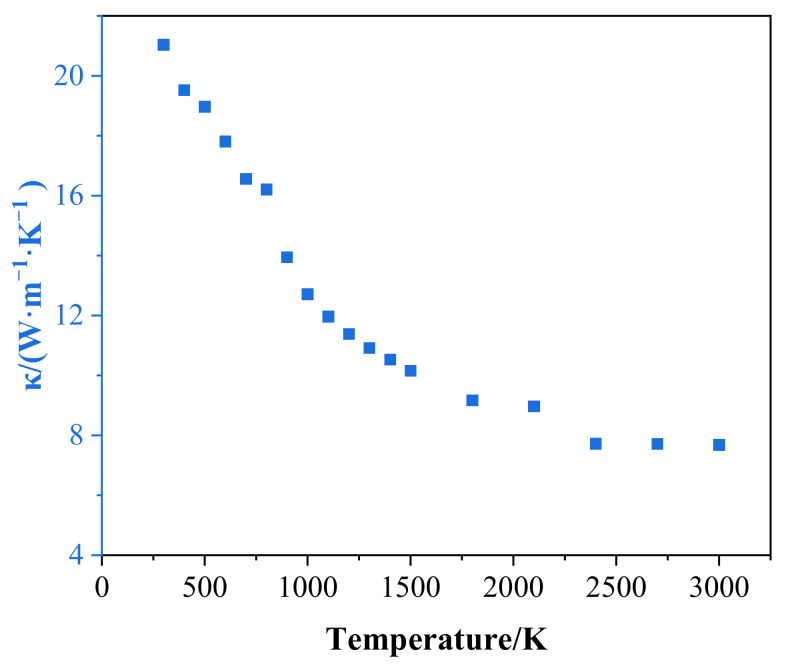
Thermal conductivity of GS-Hs in the substrate temperature range of 300 to 3000 K.

**Figure 6 nanomaterials-12-00894-f006:**
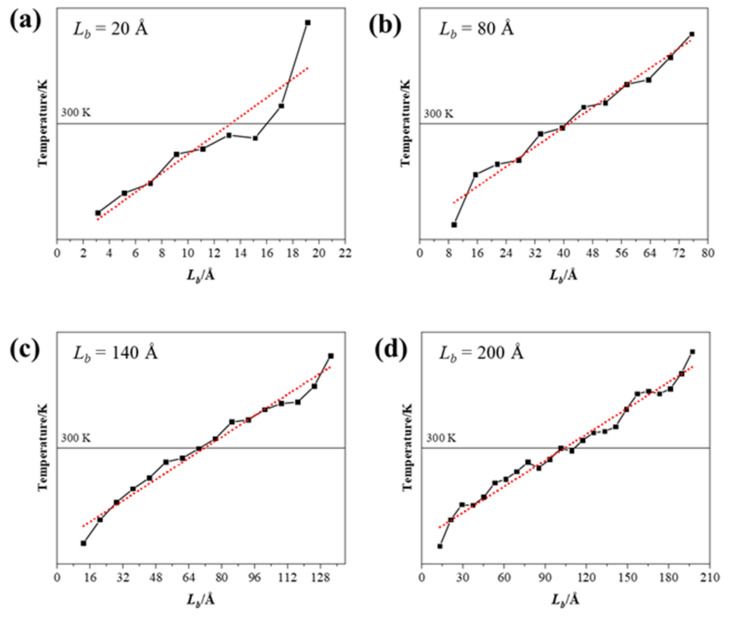
The temperature distribution and temperature gradient at system size of (**a**) 20 Å, (**b**) 40 Å, (**c**) 140 Å, and (**d**) 200 Å.

**Figure 7 nanomaterials-12-00894-f007:**
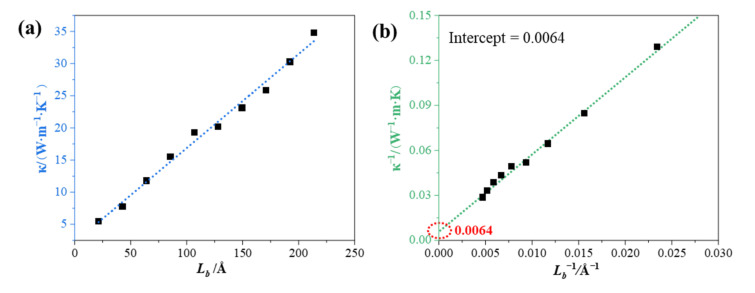
(**a**) Thermal conductivities as a function of the system size. (**b**) Inverse of thermal conductivity as a function of the inverse of the system size.

**Table 1 nanomaterials-12-00894-t001:** Lennard-Jones parameters.

Elements	ε (eV)	σ (Å)
Si-C	0.00891	3.629
Si-Si	0.01740	3.826
C-C	0.00455	3.431

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
