# Peer review of "Enhanced Thermal Transport Properties of Graphene/SiC Heterostructures on Nuclear Reactor Cladding Material: A Molecular Dynamics Insight"

_nanomaterials, 2022, doi:10.3390/nano12060894_

Round 1
Reviewer 1 Report
- The main finding and its significance should be added in the abstract.
- The differences between this study and the previous ones are not clear.
- The authors should thoroughly check the references for their accuracy and completeness. Some are without page number and some are without journal name.
Author Response
Dear Editor and Reviewers:
We sincerely appreciate the time dedicated by the reviewers on the manuscript, and thank them for all valuable comments and suggestions. Our pointwise responses are presented in blue text, whereas the reviewers’ comments are repeated verbatim below. The corresponding revisions made to the main manuscript and other files are highlighted in red.

Reviewer 2 Report
The manuscript entitled “Enhanced Thermal Transport Properties of Graphene/SiC Heterostructures on Nuclear Reactor Cladding Material: A Molecular Dynamics Insight” describes the development of a heat conducting material for nuclear reactors.
The manuscript is a weak contribution to nanomaterials. A single parameter of a composite material (thermal conductivity) is measured by the authors. The authors provide models to describe the effects that were observed. The conceptual approach is very narrow. In material development regularly more than one constraint needs to be taken into account. The focus on thermal conductivity for a single material appears to be of quite limited use. Please elaborate how the results can be applied to other material combinations. Also, the link to other properties needs to be made. The authors state the importance of structural stability of SiC materials. Then, experimental evidence ought to be provided.
Most importantly, I am concerned that the motivation concerns nuclear power applications. This does not seem to fit the current focus on sustainability. Would the materials be likewise applicable to other fields, such as high-temperature solar receivers?
Numerous issues came to my attention when reading the revised manuscript.
The authors need to give consideration to the outcome of the study. What can be learned from the study for other related materials? Is the model that is developed applicable also to other materials?
I may have missed it, but I could not locate the description on how the material was prepared and thermal conductivities have been measured.
Figure 5 shows a decrease in „Thermal conductivity of GS-Hs in the substrate temperature range of 300 to 3000 K“. The authors have used an exponential decline. Please derive, if this follows from the theory that is developed within of the study. The data would also fit with an s-shaped curve. Please elaborate.
Generally, writing ought to be much more specific. Thus, on page 7, line 218/2019 the authors state „The phonon mean free path reduction is caused by the finite size“. Please state the mean free paths. Please state the finite sizes. Precisely describe your model with regard to the processes occurring within the grain, at the grain boundary and within the matrix.
Conclusions need to be revised. The authors need to state to state clearly, what has been learned from this study. A statement that a new material has been engineered for a specific purpose is not enough.
- Figure 4 – scale bar is missing on the y-axis.
- Figure 6 – scale bar is missing on the y-axis.
- Figure 7b – units missing
Also the use of the English needs to be checked again throughout the entire manuscript. Some examples where improvements are necessary are given below.
- Page 1, line 16, Clarify what you mean with „Continuous addition“
- Page 1, line 38, check, if you want to refer to industries in plural - in my understanding this is one industry.
- Page 2, line 48/49, incomplete sentence
- Page 8, line 254, check and clarify use of the word „emphatically“
- Page 8, line 261/262, the sentence „The system size dependence of the thermal conductivity suggests the simulation results of finite systems are extrapolated to the infinite system size.“ Does not carry a meaning. Please check
- Figure 5 Check spelling of „Temperture/K“
Also the references need to be checked.
- Page number are frequently missing.
- Initials have been mixed up, as in
- J.J.o.A.P. Pop
- F.J.T.J.o.c.p. Ely
- J.I.J.o.T. Goodson
In conclusion, the study will need to be extended. The manuscript will require careful revision. Further experimental data will be needed.
Author Response

(The authors gave the same response as above.)

Reviewer 3 Report
The paper represents an additional contribution to the theoretical/numerical studies on graphene-based composites properties.
The paper is globally well written, although some points should be better defined:
1) the paper addresses the need for better materials to be used in nuclear reactor plants. However, no precise benchmark is given to compare the results and the targeted application. It is not clear if the results should be considered positive, negative, or neutral.
2) The use of Fourier law in the y-direction is not very clear. As far as I have understood, the relevant conductivity for the targeted application is in the direction perpendicular to y. Could the author clarify this aspect? it looks to me that the thermal conductivity they measure will be relevant for spreading the heat.
Author Response

(The authors gave the same response as above.)

Round 2
Reviewer 2 Report
The manuscript entitled “Enhanced Thermal Transport Properties of Graphene/SiC Heterostructures on Nuclear Reactor Cladding Material: A Molecular Dynamics Insight” describes a theoretical study on heat conducting composite material for nuclear reactors.
The manuscript has been revised. The issues pointed out were given superficial attention and some minor modifications of the manuscript have been done. Nevertheless, the manuscript is a weak contribution to nanomaterials.
Further clarification will be needed. The theoretical model will need to be based on experimental evidence. The authors need to demonstrate that this type of material actually can be made.
The following points pointed out by the reviewer on the initial manuscript have not been addressed adequately:
- A single parameter of a composite material (thermal conductivity) is measured by the authors. – Further evidence will need to be provided.
- The conceptual approach is very narrow. –Further insights will be needed.
- Please elaborate how the results can be applied to other material combinations. – Please provide evidence.
- The authors state the importance of structural stability of SiC materials. - Experimental evidence ought to be provided.
- Nuclear power applications do not fit the current focus on sustainability. – The authors state that the application is immature. So, if the material is not applicable for the intended application, why is there a need to develop a theory on this material?
- What can be learned from the study for other related materials? – Please provide evidence that the finding can be applied to other related materials.
- Is the model that is developed applicable also to other materials? – Please provide evidence.
- Figure 5 – Please show that the theory developed by the authors is applicable.
- Generally, writing ought to be much more specific. – Text has not been revised except for the parts pointed out in red; further clarification will be needed.
- Conclusions need to be revised. - What has been learned from this study?
- Figure 4 and 6, scale bar missing on the y-axis; Figure 7b, units missing – Figures will need to be updated.
- Also, the use of the English needs to be checked again throughout the entire manuscript. – This will need to be done.
- Page 7, “… The biggest size of this paper …” – please clarify
- References – Please check journal names.
Abstract:
- How can graphene layers be provided continuously? When added, the layers will form discrete steps.
- How can the interaction decrease with the distance between SiC and graphene, when the graphene is embedded in the SiC? Please clarify.
- System size – please specify.
- Meanwhile – please check choice of words
- Available – please correct spelling mistake
- Please provide evidence that the theory is applicable to other systems.
The conclusions remain the same: The scope of the study will need to be extended. Experimental data will be needed to verify the claims of the authors. The manuscript requires further careful revision. I do not recommend this manuscript for nanomaterials. Other MDPI journal focusing more on theory and physics may be more suitable.
Author Response
We sincerely appreciate the time dedicated by the reviewers on the manuscript, and thank them for all valuable comments and suggestions. Our pointwise responses are presented in blue text, whereas the reviewers’ comments are repeated verbatim below. The corresponding revisions made to the main manuscript and other files are highlighted in red.
